# X-ray Structure Refinement and Vibrational Spectroscopy of Metavauxite FeAl$_2$(PO$_4$)$_2$(OH)$_2$·8H$_2$O

**Giancarlo Della Ventura [1,2]**, **Francesco Capitelli [3]**, **Giancarlo Capitani [4]**, **Gennaro Ventruti [5,*]** and **Alessandro Monno [5]**

[1] Dipartimento di Scienze, Università Roma Tre, Largo S. L. Murialdo 1, 00146 Rome, Italy; giancarlo.dellaventura@uniroma3.it
[2] Istituto Nazionale di Fisica Nucleare, Via E. Fermi 40, I-00044 Rome, Italy
[3] Institute of Crystallography-CNR, Via Salaria Km 29.300, 00016 Monterotondo (Roma), Italy; francesco.capitelli@ic.cnr.it
[4] Dipartimento di Scienze dell'Ambiente e della Terra (DISAT), Piazza della Scienza 4, 20126 Milano, Italy; giancarlo.capitani@unimib.it
[5] Dipartimento di Scienze della Terra e Geoambientali, Università di Bari, via E. Orabona 4, 70125 Bari, Italy; alessandro.monno@uniba.it
[*] Correspondence: gennaro.ventruti@uniba.it; Tel.: +39-080-5442596

**Abstract:** In this paper, we provide a crystal-chemical investigation of metavauxite, ideally FeAl$_2$(PO$_4$)$_2$(OH)$_2$·8H$_2$O, from Llallagua (Bolivia) by using a multi-methodological approach based on EDS microchemical analysis, single crystal X-ray diffraction, and Raman and Fourier transform infrared (FTIR) spectroscopy. Our new diffraction results allowed us to locate all hydrogen atoms from the structure refinements in the monoclinic $P2_1/c$ space group. Metavauxite structure displays a complex framework consisting of a stacking of [Al(PO$_4$)$_3$(OH)(H$_2$O)$_2$]$^{7-}$ layers linked to isolated [Fe(H$_2$O)$_6$]$^{2+}$ cationic octahedral complex solely by hydrogen bonding. The hydrogen-bonding scheme was inferred from bond-valence calculations and donor-acceptor distances. Accordingly, strong hydrogen bonds, due to four coordinated H$_2$O molecules, bridge the [Fe(H$_2$O)$_6$]$^{2+}$ units to the Al/P octahedral/tetrahedral layer. The hydroxyl group, coordinated by two Al atoms, contributes to the intra-layer linkage. FTIR and Raman spectra in the high-frequency region (3700–3200 cm$^{-1}$) are very similar, and show a complex broad band consisting of several overlapping components due to the H$_2$O molecules connecting the isolated Fe(H$_2$O)$_6$ and the adjacent Al/P octahedral/tetrahedral layers. A sharp peak at 3540 cm$^{-1}$ is assigned to the stretching mode of the OH group. The patterns collected in the low-frequency region are dominated by the stretching and bending modes of the PO$_4$$^{3-}$ group and the metal-oxygen polyhedra.

**Keywords:** metavauxite; phosphate; single-crystal X-ray structure refinement; hydrogen bonding network; FTIR and Raman spectroscopy

## 1. Introduction

Metavauxite, ideally FeAl$_2$(PO$_4$)$_2$(OH)$_2$·8H$_2$O, is a rare supergene secondary phosphate found in oxidized zones of tin mines at Llallagua, Bolivia. It occurs associated with other hydrated aluminum phosphates, such as paravauxite FeAl$_2$(PO$_4$)$_2$(OH)$_2$·8H$_2$O [1–3], vauxite FeAl$_2$(PO$_4$)$_2$(OH)$_2$·6H$_2$O [4–6], and wavellite Al$_3$(PO$_4$)$_2$(OH,F)$_3$·5H$_2$O [7,8].

Due to its rarity, spectroscopic and structural investigations on metavauxite are very limited. *De facto*, the only available structural study is the original structure model published long ago by Baur and Rama Rao [9]. The described structure was, however, lacking a direct description of the hydrogen

bonding network, due to undetermined H positions. Later, accurate X-ray powder diffraction data were reported by [10], providing improved information for a correct identification of this phosphate. Few structure details of metavauxite were also discussed by [11] within a general crystal-chemical model addressing the water of hydration in minerals. The present study aims at providing: (a) A new X-ray single-crystal structure refinement based on new X-ray data allowing a significant improvement of the structure model, (b) the hydrogen-bonding scheme from the experimental determination of hydrogen positions, and (c) FTIR and Raman data, which are lacking in the literature for this mineral.

## 2. Studied Sample and Experimental Methods

Various rare and historically precious phosphates, belonging to the *Alberto Pelloux* collection, are kept at the Earth Science Museum of the University of Bari Aldo Moro (Italy). The metavauxite sample examined here (Figure 1) was collected during the fourth mineralogical expedition in 1925 at Llallagua (Bolivia) by S.G. Gordon, who gave the first description of its occurrence [12,13]. The sample of the Pelloux Collection belongs to the same batch, later studied by [9].

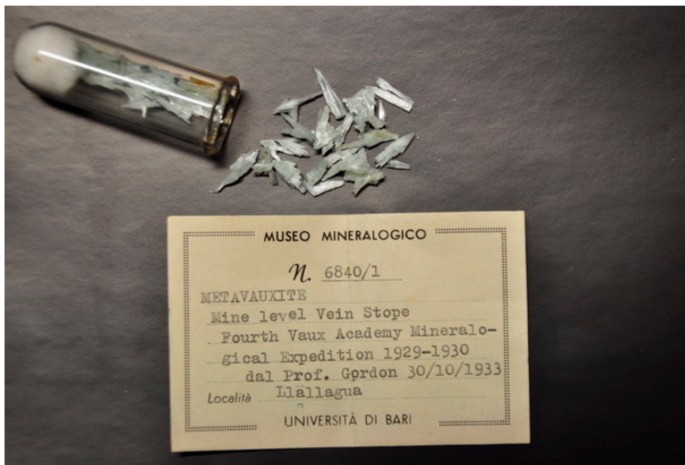

**Figure 1.** Metavauxite sample collected during the fourth mineralogical expedition (1925) by S.G. Gordon [12,13].

The chemical composition was determined on a fragment of the same crystal used for X-ray single-crystal structural refinement with a Tescan VEGA TS 5136XM equipped with an EDAX GENESIS 4000XMS EDS system. Due to the difficulties in polishing such cleavable and fragile crystals and the expected water loss and elemental diffusion under intense electron beam, EDS was preferred to WDS because of less demanding preparation requirements and much lower operational beam current (190 pA vs. more than 5 nA for EDS and WDS, respectively). To further reduce any possible elemental diffusion during acquisition, we performed selected area analysis instead of point analysis. These were acquired at 20 keV on flat cleavage fragments, and quantified using standards. The following minerals/X-ray lines were employed: Apatite/P-K$\alpha$, biotite/Al-K$\alpha$ and Fe-K$\alpha$, and arfvedsonite/Mn-K$\alpha$. Data reduction used the ZAF correction method [14].

X-ray data were collected using a Bruker AXS X8 automated diffractometer, equipped with an APEXII CCD detector, using graphite-monochromatized Mo$K\alpha$ radiation ($\lambda$ = 0.71069 Å). Operating conditions were: 50 kV, 30 mA. Three sets of 12 frames were acquired with 0.5° $\varphi$ rotation for the initial unit-cell determinations. The collection strategy was optimized by the Apex program suite [15]; the intensities of reflections in the entire Ewald sphere ($\pm h$, $\pm k$, $\pm l$) were recorded by combining $\omega$ and $\varphi$ rotation sets with a 0.5° scan width and exposure times from 10 to 30 s/frame. The integration of reflection intensities and the correction for Lorentz-polarization (Lp), background effects, and scale variation were performed by the SAINT package [16]. A semi-empirical absorption correction was applied using the SADABS software [17]. The final unit-cell parameters were obtained from the xyz

centroids of the all integrated reflections yielding the following unit-cell constants: $a$ = 10.2449(5) Å, $b$ = 9.5867(5) Å, $c$ = 6.9626(3) Å, $\beta$ = 97.889(3)° (Table 1). Data analysis, preliminarily carried out with Xprep software [18], confirmed the space group $P2_1/c$. The structural model was anisotropically refined by a full-matrix least-squares method based on $F^2$ using CRYSTALS [19] based on the ideal formula $FeAl_2(PO_4)_2(OH)_2 \cdot 8H_2O$. The final stages of the refinement did not indicate the need of an extinction correction. The hydrogen atoms of $H_2O$ molecules and the hydroxyl groups were located by successive difference Fourier maps and refined isotropically. Lastly, since electron density maps revealed short O-H distances, as observed in other hydrous phosphate phases (wavellite [8], vauxite [5]), these were restrained to a target value $d$ = 0.89(4) Å, while isotropic atomic displacement parameters of hydrogen atoms were fixed as 1.2*Ueq of the parent oxygen atom [5,20]. This is a common problem when using X-rays to locate the H atoms from the electron density maps that provide only bond orientations and the H positions need to be restrained. The final cycle of least-squares refinement included 133 parameters, with CRYSTALS weighting scheme applied: $w^{-1} = [s^2(F)^2 + (0.10)P]$ where $P = p(6)*\max(Fo^2,0) + (1-p(6))Fc^2$; final $R$ indices were $R_1$ = 0.0381 and $wR_2$ = 0.0381 for I > 2$\sigma$(I), while for all data they were $R_1$ = 0.0759 and $wR_2$ = 0.1369.

**Table 1.** Crystal data and structure refinement for metavauxite from Lallagua, Bolivia.

| Empirical Formula | $H_{18}Al_2FeO_{18}P_2$ |
|---|---|
| Formula weight | 477.89 |
| Temperature (K) | 293(2) |
| Wavelength (Å) | 0.71069 |
| Crystal system; space group | Monoclinic; $P2_1/c$ |
| $a$, $b$, $c$ (Å) | 10.2449(5), 9.5867(5), 6.9626(3) |
| $\beta$ (°) | 97.889(3) |
| $V$ (Å$^3$) | 677.36(6) |
| $Z$, $\rho_{calc.}$ (g·cm$^{-3}$) | 2, 2.343 |
| $\mu$ (mm$^{-1}$) | 1.584 |
| F(000) | 488 |
| Crystal size (mm) | $0.35 \times 0.25 \times 0.25$ |
| Shape, Color | prismatic, colorless |
| $\theta$ range for data collection | 3.64 to 36.37 deg. |
| Limiting indices | $-17 \le h \le 17, -15 \le k \le 15, -8 \le l \le 11$ |
| Refl. collected/unique | 15071/3273 [$R_{int}$ = 0.084] |
| Completeness | 99.5% |
| Max. and min. transmission | 0.6227 and 0.7471 |
| Refinement method [a] | FMLS on $F^2$ |
| Data/restraints/parameters | 2018/9/133 |
| GOF | 0.780 |
| Final $R$ indices [I>3$\sigma$(I)] | $R_1$ = 0.0381, $wR_2$ = 0.1005 |
| R indices (all data) | $R_1$ = 0.0759, $wR_2$ = 0.1369 |
| Larg. diff. peak/hole (e·Å$^{-3}$) | 0.84/−0.68 |

[a] Wincrystal weighting scheme applied $w^{-1} = [s^2(F)^2 + (0.10)P]$ where $P = p(6)*\max(Fo^2,0) + (1 - p(6))Fc^2$.

Structure drawings were done by using DIAMOND [21], and VESTA [22]. Further details of the crystal structure investigation can be obtained from the Fachinformationszentrum Karlsruhe, 76344 Eggenstein-Leopoldshafen, Germany, (Fax: (+49-7247-808-666; e-mail: crysdata@fiz-karlsruhe.de) on quoting the depository number CSD-429794. The list of $F_o/F_c$ data is available from the authors up to one year after the publication of the work. Crystal structure refinement data are reported in Table 1; fractional atomic coordinates and equivalent isotropic displacement parameters are reported in Table 2; selected bond lengths are reported in Table 3.

**Table 2.** Fractional atomic coordinates, Wyckoff notation, and equivalent isotropic displacement parameters ($\text{Å}^2$) for metavauxite from Llallagua, Bolivia.

| Atom | Site | x/a | y/b | z/c | $U_{eq}$ |
|---|---|---|---|---|---|
| Fe1 | 2a | 0 | 0.5 | 0.5 | 0.0132(2) |
| Al1 | 4e | 0.51123(7) | 0.25360(7) | 0.12761(9) | 0.0083(3) |
| P1 | 4e | 0.32683(6) | 0.53757(6) | 0.07485(8) | 0.0078(2) |
| O1 | 4e | 0.39385(17) | 0.40061(17) | 0.0272(2) | 0.0113(7) |
| O2 | 4e | 0.61134(18) | 0.09723(18) | 0.2301(2) | 0.0128(7) |
| O3 | 4e | 0.66313(16) | 0.35767(18) | 0.0919(2) | 0.0105(7) |
| O4 | 4e | 0.18026(17) | 0.50637(19) | 0.0804(3) | 0.0135(7) |
| O5 | 4e | 0.50787(17) | 0.32012(18) | 0.3784(2) | 0.0108(7) |
| O6 | 4e | 0.35714(19) | 0.1391(2) | 0.1495(3) | 0.0163(8) |
| O7 | 4e | 0.03165(19) | 0.3990(2) | 0.7827(3) | 0.0167(8) |
| O8 | 4e | −0.12527(19) | 0.6415(2) | 0.6185(3) | 0.0174(8) |
| O9 | 4e | 0.1659(2) | 0.6331(2) | 0.5704(3) | 0.0236(10) |
| H51 | 4e | 0.495(3) | 0.405(3) | 0.380(5) | 0.0130 |
| H61 | 4e | 0.281(3) | 0.161(4) | 0.091(5) | 0.0200 |
| H62 | 4e | 0.348(4) | 0.120(4) | 0.271(4) | 0.0200 |
| H71 | 4e | 0.087(3) | 0.438(4) | 0.867(5) | 0.0200 |
| H72 | 4e | −0.046(3) | 0.411(4) | 0.822(5) | 0.0200 |
| H81 | 4e | −0.155(4) | 0.606(4) | 0.727(5) | 0.0210 |
| H82 | 4e | −0.088(4) | 0.719(3) | 0.662(5) | 0.0210 |
| H91 | 4e | 0.218(4) | 0.635(4) | 0.675(5) | 0.0290 |
| H92 | 4e | 0.213(4) | 0. 668(4) | 0.493(5) | 0.0290 |

**Table 3.** Bond lengths ($\text{Å}$) and selected angles (°) for metavauxite from Llallagua, Bolivia.

| | | | | | |
|---|---|---|---|---|---|
| Fe1–$H_2$O7 [I] | 2.1776(19) | Al1–O1 | 1.9211(17) | P1–O3 [III] | 1.5494(17) |
| Fe1–$H_2$O7 [II] | 2.1776(19) | Al1–O2 | 1.8994(18) | P1–O4 | 1.5367(18) |
| Fe1–$H_2$O8 [I] | 2.1113(19) | Al1–O3 | 1.8934(18) | P1–O2 [IV] | 1.5285(17) |
| Fe1–$H_2$O8 [II] | 2.1113(19) | Al1–O5H | 1.8636(16) | P1–O1 | 1.5388(18) |
| Fe1–$H_2$O9 [I] | 2.128(2) | Al1–O5[I]H | 1.8695(17) | <P1–O> | *1.5384* |
| Fe1–$H_2$O9 [II] | 2.128(2) | Al1–$H_2$O6 | 1.9459(19) | | |
| <Fe1–O> | *2.139* | <Al1–O> | *1.899* | | |

(I) x, 0.5 − y, −0.5 + z; (II) −x, −0.5 + y, 0.5 − z; (III) −x + 1, −y + 1, −z; (IV) 1 − x, 0.5 + y, 0.5 − z.

Powder FTIR spectra in the MIR (medium infrared) range were collected using a Bruker V70 FTIR spectrometer equipped with a DTGS (deuterated triglycine sulfate) detector and an extended-range KBr beamsplitter; the nominal resolution was 4 $\text{cm}^{-1}$ and 256 scans were averaged for both sample and background. The sample was prepared as a KBr disk by mixing about 1 mg of metavauxite powder within 200 mg of KBr. The spectrum at 77K was collected by using a cryostat installed on the same instrument, under the same experimental conditions.

Single-crystal Raman spectra were obtained with a confocal Horiba Jobin Yvon Labram HR Evolution spectrometer equipped with an Olympus optical microscope, an ultra-low frequency (ULF) filter, and a multichannel air-cooled CCD detector. Raman spectra over a range of 10–3800 $\text{cm}^{-1}$ were excited with the He–Ne 632.8 nm line. A 50× long distance objective was used and the laser power was decreased to avoid potential local heating effects due to heavy light absorption. The sample was inspected optically for any laser damage, and none was observed. The wavenumber accuracy was ±0.5 $\text{cm}^{-1}$, with a spectral resolution <1 $\text{cm}^{-1}$. Each spectrum was accumulated three times with an integration time of 60 s to improve the signal to noise ratio. The Raman peak position of a silicon substrate (520.5 $\text{cm}^{-1}$) was used to calibrate spectral frequency.

## 3. Results and Discussion

### 3.1. Microchemical Composition of the Studied Sample

The microchemical data of the investigated metavauxite gave the following oxide wt%, average of eight analyses: $P_2O_5$ = 30.73, $Al_2O_3$ = 22.63, MnO = 0.16, FeO = 14.00, $H_2O$ (calc) = 32.47. No other elements were detected in the EDS spectrum, confirming the sample to be very close to the ideal composition. The crystal-chemical formula, calculated on the basis of nine oxygen atoms and ideal $H_2O$ content, is $Fe_{0.91}Mn_{0.01}Al_{2.05}P_{2.00}O_8(OH)_2 \cdot 8H_2O$.

### 3.2. X-ray Structure Refinement

#### 3.2.1. Description of the Structure

Refined atomic positions are reported in Table 2, while bond distances are reported in Table 3. All atoms occur on general positions, with the exception of Fe1 lying on 2*a* special site (Table 2); Figure 2 displays local cationic and anionic environments within the structure.

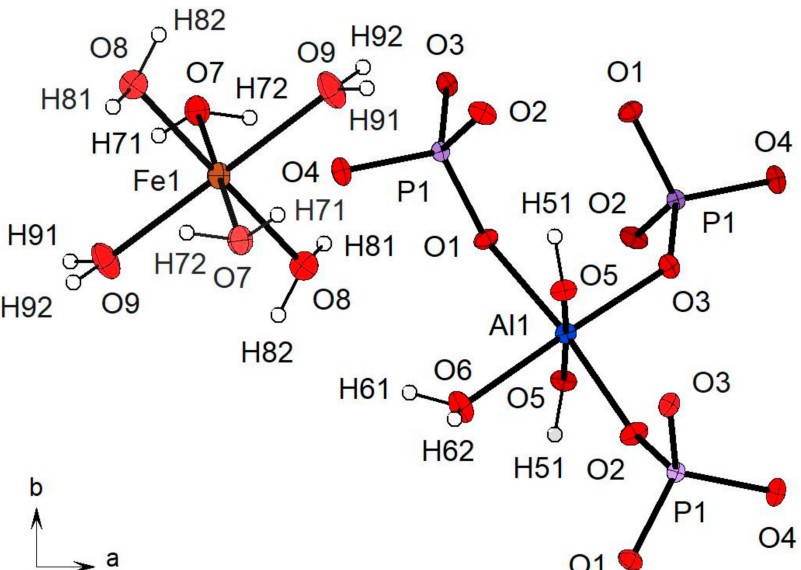

**Figure 2.** Local cationic and anionic environments for the investigated metavauxite from Llallagua. Displacement ellipsoids are drawn at 50% probability level. Figure drawn by DIAMOND [21].

Owing to its site symmetry, $Fe^{2+}$ is involved in a regular octahedral $Fe(H_2O)_6$ coordination (Figure 2 and Table 2), with bond distances (x2) $Fe1–H_2O8$ and $Fe1–H_2O7$, respectively 2.1113(19) and 2.1776(19) Å. The $Fe1–H_2O8$ and $Fe1–H_2O7$ define the equatorial plane of the octahedron while the O9 and its *trans*-related anion, at distance (x2) $Fe1–H_2O9$ = 2.128(2) Å (Table 3) define the apexes of the octahedron. Internal angles display the largest deviations from the ideal value of 90° at O7–Fe1–O8 = 87.46(7)°. All $Fe–H_2O$ bond distances are in good agreement with those observed in natural Fe phosphates [23,24].

Aluminum is characterized by a $AlO_3(OH)_2(H_2O)$ octahedral environment, with bond distances ranging from 1.8636(16) (Al1–O5H) to 1.9211(6) Å (Al1–$H_2O6$) (Table 3); owing to the general position of coordinating central cation, this octahedron appears rather distorted; the equatorial plane can be interpreted as the one formed by O1, O2, O3, and O6 atoms, with the shortest Al–eq. plane distance = 0.030 Å. Internal angles are very close to 90° (largest deviation at O2–Al1–O5 89.46(7)°), while apical positions are represented by O5 and its symmetry related (Figure 2), being the angle O5–Al1–O5 of 176.96(9). Every Al cation is further coordinating three $PO_4^{-3}$ groups (Figure 2) by sharing vertices in the equatorial plane, with significant consequences in three-dimensional framework making-up, as

later focused. Al–O bond distances range from 1.8636(16) up to 1.9459(19) Å, in good agreement with analogue distances commonly observed in natural phosphates [24,25].

Phosphorous atom displays tetrahedral coordination (Figure 2), with bond distances ranging from 1.5285(17) (P1–O2) to 1.5494(17) Å (P1–O3) (Table 3). Refined P–O bond distances are in good agreement with those observed in natural Al phosphates [8,26,27], and with values reported by [24]. The ($PO_4$) tetrahedron show internal angles slightly larger than the ideal value of 104°, displaying the largest deviation at O1–P1–O1 = 111.18(8) and at O2–P1–O3 = 111.60(9)°.

The structure of metavauxite is based on infinite chains of vertex-sharing $AlO_3(OH)_2(H_2O)$ octahedra running parallel to the **c** axis, being O5 the shared vertex oxygen atom (Figure 3a); the $Al^{3+}$ chains are interconnected by three ($PO_4$) groups via sharing three oxygen atoms of the equatorial plane of Al polyhedron (Figure 2). In this way, infinite layers parallel to (010) are made up (Figure 3a). These layers are, in turn, connected to isolated $Fe(H_2O)_6$ octahedra, also arranged to form layers parallel to (010) via strong hydrogen bonds, thus building up the three-dimensional framework of metavauxite (Figure 3b).

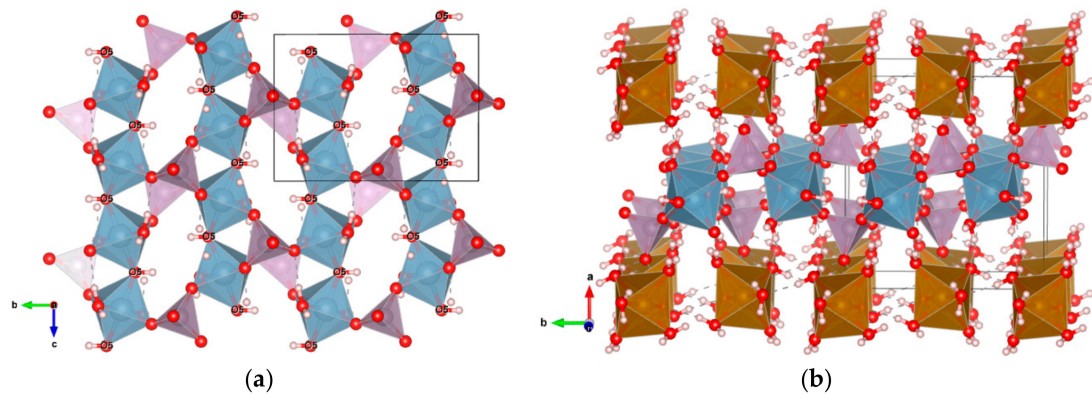

(**a**)                                                           (**b**)

**Figure 3.** (**a**) View of the b–c plane of octahedra along the **a** direction showing the Al1–P1 connections. (**b**) Projection the metavauxite structure along the **c** direction showing the polyhedral layering. The Al1 octahedra are in blue, the P1 tetrahedra in pink, the oxygen atoms in red, the Fe1 octahedra in brown, and the H atoms in white. The unit cell is indicated. Figure drawn by VESTA [22].

### 3.2.2. Polyhedral Distortion

The polyhedral distortions can be quantified by using the distortion index (DI) defined as [(Cation-$O_{max}$)-(Cation-$O_{min}$)]/<Cation-O> [28]. The studied metavauxite shows distortion values of 0.031 and 0.043 for, respectively, Fe1 and Al1 octahedra, and tetrahedral distortion of 0.014 for the P1 tetrahedron. The DI for $FeO_6$ octahedra ($Fe^{2+}$ ionic radius = 0.78 Å [29]) is compared in Figure 4a with the data from the most common $Fe^{2+}$ phosphates as found in the ICSD [24]. The plot shows a positive correlation between DI and <Fe–O> distance. The DI for $AlO_6$ octahedra, calculated for the most common natural Al phosphates, is plotted in Figure 4b. The low range of values is mainly due to the small $Al^{3+}$ ionic radius (0.54 Å [29]), in comparison with $Fe^{2+}$ or other cations usually present in phosphates (see for example [30]). Inspection of Figure 4b shows that, differently from what observed for $Fe^{2+}$, the distortion index is not correlated with the <Al–O> distance. The mean bond distance <Al1–O> in metavauxite (1.8988(18) Å), as recently observed in wavellite [8] and vauxite [5], is slightly shorter in comparison with other natural Al phosphates from the ICSD [24] or from our previous investigations of whiteite $CaFeMgAl_2(PO_4)_4(OH)_2 \cdot 8H_2O$ [26] and augelite $Al_2(PO_4)(OH)_3$ [27].

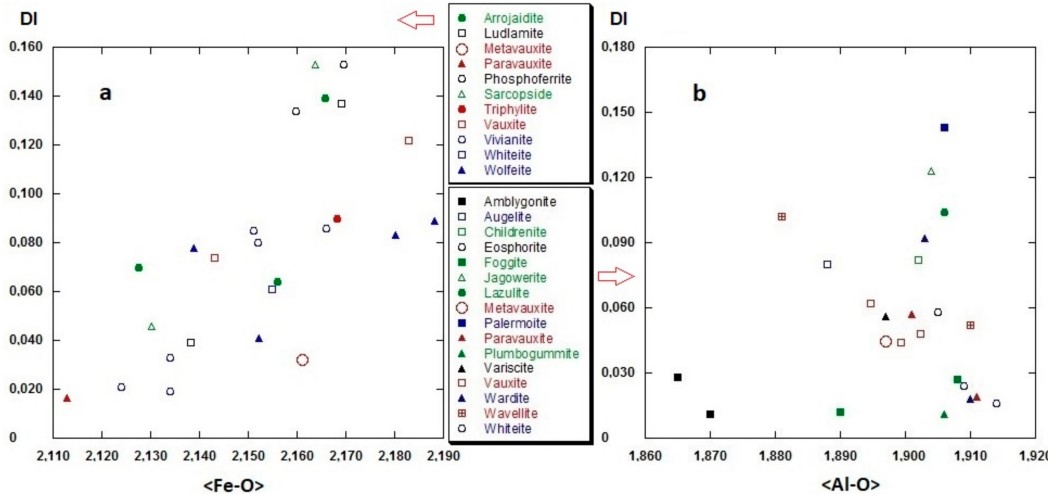

**Figure 4.** Distortion index (DI) [28] trends for $FeO_6$ octahedra for selected $Fe^{2+}$ phosphates (**a**) from [23], and for for $AlO_6$ octahedra for selected Al phosphates (**b**) from the literature [24]. Paravauxite: [3]. Vauxite: [5]. Whiteite: [26]. Wavellite: [8]. Augelite: [27]. Metavauxite: This work.

### 3.2.3. Hydrogen Bonding Network

During the present work, due to the higher resolving power of the used X-ray diffractometer, we were able to locate all H atoms unambiguously, allowing a complete description of the complex H-bonding scheme (Table 4), not provided in the original work of [9]. Basically, all H-bonding interactions join the chain of discrete $Fe(H_2O)_6$ octahedra with the layer consisting of Al octahedra and $PO_4^{3-}$ tetrahedra, the only exception being O8–H82 . . . O7$^{VI}$, (D . . . A = 2.703(3) Å) linking each other the single Fe octahedra (Figure 5). The strongest interactions are those formed by O7, displaying D . . . A distances of 2.609(3) and 2.648(3) Å, for O7–H71 . . . O4$^{IV}$ and O7–H72 . . . O4$^V$ systems, respectively. Both the interactions concern the Fe octahedral chain, linking every single $[Fe(H_2O)_6]^{2+}$ unit with $PO_4^{3-}$ tetrahedra respectively up and down along **a**, normal to the direction of the chain. The $H_2O8$ and $H_2O9$ molecules of $[Fe(H_2O)_6]^{2+}$ units are further connected to the $PO_4^{3-}$ tetrahedra, acting as donors in O8–H81 . . . O4$^V$, O9–H91 . . . O3$^{VII}$, and O9–H92 . . . O3$^I$ interactions, displaying D . . . A distances of, respectively, 2.654(3), 2.734(2), and 3.086(3) Å. Besides, $H_2O8$ molecule of Fe octahedron, is connected also with $AlO_3(OH)_2(H_2O)$ environment, acting in this case, as an acceptor in the O6–H61 . . . O8$^{II}$ interaction, displaying a D . . . A distance of 2.812(3) Å. Lastly, the hydroxyl group connecting all Al1 octahedra to form the aluminum chain extended along **c** (Figure 3a), is involved in a weak O5–H51 . . . O2$^I$ interaction (D . . . A = 2.978(2) Å) connecting the Al1 octahedron with the ($PO_4$) units.

**Table 4.** Hydrogen bonds (Å, °) for metavauxite from Llallagua, Bolivia.

| D-H . . . A | D-H | H . . . A | D . . . A | D-H-A |
|---|---|---|---|---|
| O5–H51 . . . O2 $^I$ | 0.83(3) | 2.22(3) | 2.978(2) | 152(3) |
| O6–H61 . . . O8 $^{II}$ | 0.86(3) | 2.02(3) | 2.812(3) | 154(4) |
| O6–H62 . . . O1 $^{III}$ | 0.88(3) | 1.79(3) | 2.632(2) | 158(4) |
| O7–H71 . . . O4 $^{IV}$ | 0.84(3) | 1.78(3) | 2.609(3) | 167(4) |
| O7–H72 . . . O4 $^V$ | 0.88(3) | 1.80(3) | 2.648(3) | 161(4) |
| O8–H81 . . . O4 $^V$ | 0.92(3) | 1.76(3) | 2.654(3) | 161(4) |
| O8–H82 . . . O7 $^{VI}$ | 0.87(3) | 1.85(3) | 2.703(3) | 169(4) |
| O9–H91 . . . O3 $^{VII}$ | 0.84(3) | 1.89(3) | 2.734(2) | 179(4) |
| O9–H92 . . . O3 $^I$ | 0.85(3) | 2.33(3) | 3.086(3) | 148(4) |

(I) −x + 1, y + 0.5, 0.5 − z; (II) −x, −0.5 + y, 0.5 − z; (III) x, 0.5 − y, 0.5 + z; (IV) x, y, z + 1; (V) −x, 1 − y, 1 − z; (VI) −x, 0.5 + y, 1.5 − z; (VII) 1 − x, 1 − y, 1 − z.

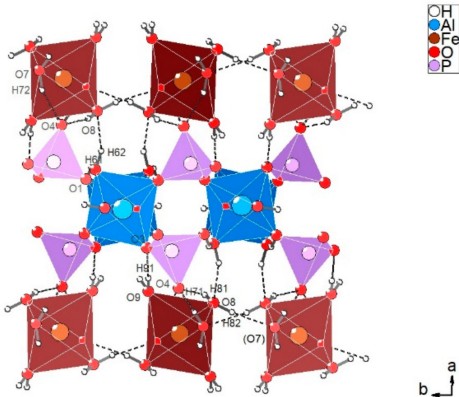

**Figure 5.** Hydrogen bonding network of metavauxite, viewed down **c**. For the sake of clarity, only interactions H . . . Acceptor up to 2.10 Å are shown. Figure drawn by DIAMOND [21].

Analysis of bond-valence sum (BVS) [31] reveals a slight overbonding for Al and a weak underbonding for the P atoms (Table 5). The bond-valence sums [32] for all oxygen atoms obtained after incorporating the effect of hydrogen bonding interactions, calculated from the O . . . O distances [33], are satisfactory. Slight anomalies are observed for O2 that are weakly underbonded, and O3 and O8 that are slightly overbonded. The reasons of such BVS perturbations can be found in the complex topology of the structure, where the same oxygen is shared by different atoms; see, for example, the case of $H_2O8$ molecule, whose BVS value reflects the two strong O–H . . . O interactions, in which this $H_2O$ molecule is involved (Table 4). The low BVS values calculated for the O6, O7, O8, and O9 oxygen atoms (<0.5 valence units, v.u.) confirm that these can be interpreted as $H_2O$ molecules (Table 5) while the BVS of about 1.0 for O5 oxygen is consistent with an OH group. All oxygen atoms of the $PO_4$ tetrahedra (O1, O2, O3, and O4), receive a BVS significantly lower than 2.0; these are, thus, recognized as hydrogen-bond acceptors. In particular, the strongly underbonded oxygen O4, coordinated just by the P cation, is connected via hydrogen bonds with $H_2O7$ and $H_2O8$ molecules.

**Table 5.** Calculated bond-valence sum [31] for metavauxite from Llallagua, Bolivia. Values expressed in valence units.

|  | O1 | O2 | O3 | O4 | O5 | O6 | O7 | O8 | O9 | Σ |
|---|---|---|---|---|---|---|---|---|---|---|
| Fe1 |  |  |  |  |  |  | $0.302_{\times2}$ | $0.361_{\times2}$ | $0.345_{\times2}$ | 2.016 |
| Al1 | 0.482 | 0.511 | 0.519 |  | 0.563 | 0.451 |  |  |  | 3.080 |
|  |  |  |  |  | 0.554 |  |  |  |  |  |
| P1 | 1.235 | 1.270 | 1.200 | 1.242 |  |  |  |  |  | 4.947 |
| Σv |  |  |  |  |  |  |  |  |  |  |
| H51 |  | 0.135 |  |  | 0.865 |  |  |  |  | 1.000 |
| H61 |  |  |  |  |  | 0.821 |  | 0.179 |  | 1.000 |
| H62 | 0.265 |  |  |  |  | 0.735 |  |  |  | 1.000 |
| H71 |  |  |  | 0.280 |  |  | 0.720 |  |  | 1.000 |
| H72 |  |  |  | 0.255 |  |  | 0.745 |  |  | 1.000 |
| H81 |  |  |  | 0.255 |  |  |  | 0.745 |  | 1.000 |
| H82 |  |  |  |  |  |  | 0.225 | 0.775 |  | 1.000 |
| H91 |  |  | 0.210 |  |  |  |  |  | 0.790 | 1.000 |
| $\Sigma v_H$ | 1.982 | 1.916 | 2.045 | 2.032 | 1.982 | 2.007 | 1.992 | 2.060 | 2.019 |  |

### 3.2.4. Related Phases

Metavauxite is the rarest of the related vauxite family members, paravauxite [2] and vauxite [6]. From a structural point of view all these minerals belong to a large group of phosphate minerals containing laueite-type layers, i.e., layers consisting of chains of trans-corner-sharing octahedra interlinked by $(PO_4)^{3-}$ tetrahedra. However, while metavauxite and vauxite have a different

topology, metavauxite and paravauxite are based on the same infinite chain of vertex-linked oxygen Al-octahedra. Indeed, the metavauxite structure has the same topology of the strunzite $Mn(H_2O)_4[Fe(PO_4)_2(OH)H_2O]_2$ [34] and pseudolaueite $MnFe_2(PO_4)_2(OH)_2 \cdot 8H_2O$ [35] frameworks, as outlined by the graph representation showing the polyhedral connectivity [36]. All these structures are based on identical $[M(PO_4)\phi_3]$ (with $\phi$ = O,OH,H$_2$O) chains cross-linked by PO$_4$ tetrahedra to form $[M^{2+,3+}(PO_4)(OH)(H_2O)]$ sheet. In metavauxite, $[Al(PO_4)(OH)(H_2O)]$ sheets are stacked along the **a** direction and are linked through isolated Fe(H$_2$O)$_6$ octahedra by hydrogen bonds from H$_2$O molecules coordinated by Fe and Al cations. In strunzite and pseudolaueite $[Fe^{3+}(PO_4)(OH)(H_2O)]$, sheets are joined along the *a* direction both by direct vertex-sharing of tetrahedra and octahedra, and hydrogen bonds from H$_2$O molecules belonging both to the sheet and the divalent cation. Different from the strunzite and metavauxite 2D structural unit, the sequence of orientation of the up and down arrangement of tetrahedral apices in adjacent chains is inverted in the pseudolaueite sheet.

## 3.3. Infrared and Raman Spectroscopy

### 3.3.1. The H$_2$O/OH Bands

Single-crystal Raman spectra collected in the principal H$_2$O-stretching region (3800–2000 cm$^{-1}$) for different orientation of the laser beam with respect to the crystal orientation are compared in Figure 6a. In particular, the spectra were collected on a platy cleavage fragment, with the laser oriented either parallel or perpendicular to the elongation of the plate (top and middle), or by orienting the plate on one edge (bottom). As described above, the structure of metavauxite is strongly layered (Figure 3b), and thus shows a perfect cleavage, however never reported in the description of its physical properties (see, for example, the pertinent mindat.org database entry [37]). Considering the structural arrangement, the observed cleavage is expected to be parallel to the *b–c* plane (Figure 3). Spectra collected by rotating the plate 90 degrees show a strong pleochroism of the Raman signal in the H$_2$O-stretching region (Figure 6a). All spectra consist of three well-defined peaks at 3540, 3496, and 3425 cm$^{-1}$ and a broad scattering extending from 3400 to 2000 cm$^{-1}$, which is clearly due to the contribution of several overlapping components; some of these are resolved at 3118 and 3010 cm$^{-1}$ (Figure 6a), while an evident shoulder is apparent around 3190 cm$^{-1}$. Rotation of the crystal with respect to the laser beam is correlated with a change in intensity of all components, which is particularly evident for the higher-frequency peak at 3540 cm$^{-1}$, whose scattering increases by one order of magnitude when the beam is normal to the plate elongation (Figure 6a, top).

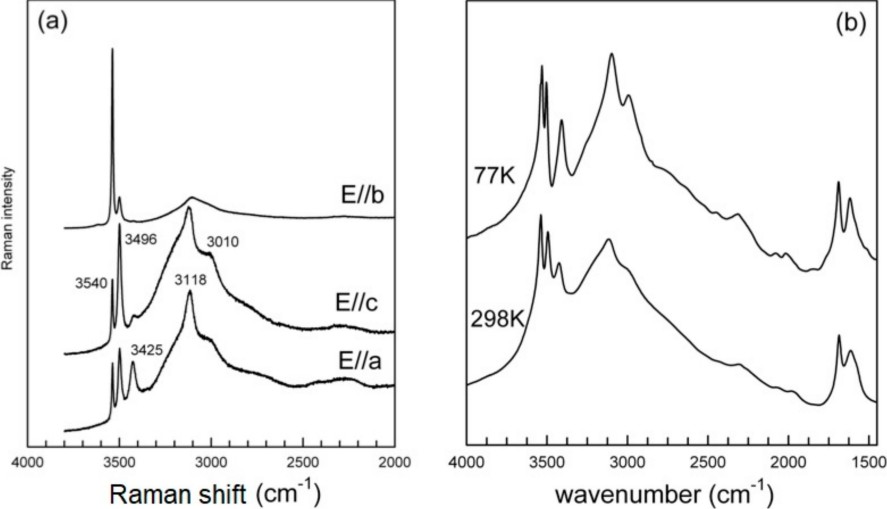

**Figure 6.** (**a**) Raman spectra in the H$_2$O-stretching region collected for different orientations of the laser beam with respect the cleavage fragment. (**b**) The 77K FTIR powder spectrum compared with the 298K FTIR spectrum.

The powder FTIR pattern (Figure 6b) collected at room-T (298K) is almost identical to the Raman spectrum and shows three well-resolved peaks on the higher-frequency side of the broad absorption, with wavenumbers coincident (within 1–2 cm$^{-1}$) to the corresponding Raman peaks (Figure 6b and Table 6).

**Table 6.** Positions (wavenumber, cm$^{-1}$) and proposed assignment for peaks observed in the FTIR and Raman spectra of metavauxite. The 77K FTIR data are given in parentheses.

| Infrared | Raman | Assignment |
|---|---|---|
| 3494 (3502), 3424 (3410), 3115 (3097), 3000 (2992) | 3496, 3425, 3188, 3010 | $\nu_3(H_2O)$, $2\nu_2(H_2O)$ |
| 1683 (1688), 1612 (1616) | 1676, 1698 | $\nu_2(H_2O)$ |
| 1142, 1084, 1049 | 1159, 1102, 1063 | $\nu_3(PO_4)$ |
| 997 | 992 | $\nu_1(PO_4)$ |
| 743, 662, 621, 603, 550, 495, 478, 427, 400 | 656, 642, 575, 495, 471, 408 | $\nu_4(PO_4)$ |
| 354, 327, 294 | 353, 303, 285, 248, 246, 230, 177, 163, 150, 136, 120, 100, 92 | $\nu_2(PO_4)/\nu(Fe\text{-}O,OH)$ |

In the $H_2O$ bending ($\nu_2$) region (Figure 6b) two peaks at 1687 and 1614 cm$^{-1}$ are present. At 77K (Figure 6b), there is a slight shift (Table 6) and a sharpening of all bands that are, thus, better resolved; see, in particular, the peak around 3010 cm$^{-1}$; no new components, notably bands due to ice formation, are observed.

In the metavauxite structure, there are four independent $H_2O$ molecules and one OH group (Table 2); considering that the $H_2O$ molecules are all involved in hydrogen bonds of different strengths with the surrounding oxygen atoms (Table 4), up to eight stretching bands originating from the single O–H bonds in the $H_2O$ molecules are ideally expected. In addition, the O–H stretching mode of the hydroxyl groups, and the first overtone of the $H_2O$ bendings ($2\nu_2$) also occur in this range, thus contributing to the complexity of the pattern (Figure 6). Decomposition of the broad absorption into different bands involves significant constraints to be introduced. However, the observed band wavenumbers, based on the empirical $O_{donor}\ldots O_{acceptor}$ distance–frequency correlation of Libowitzky [38], are all compatible with the bond distances reported in Table 4. In particular, the sharp peak at 3540 cm$^{-1}$ can be assigned to the O5–H5 hydroxyl group; the calculated wavenumber is, in fact, exactly coincident with the $O_{donor}$–$O_{acceptor}$ distance (2.978 A) observed via the structure refinement (Table 4). The local configuration of this O–H group is shown in Figure 7. The O5 oxygen is directly bonded to two nearest-neighbor Al cations and to the H51 hydrogen; this arrangement can be expressed as Al1Al1–O5 . . . H51. If the H51 hydrogen were free from any interaction with the next-nearest-neighbor surrounding oxygens, the vibrational frequency would be around 3670 cm$^{-1}$. This is the case of vauxite, where the OH-stretching mode ($\nu_{OH}$) is close to 3660 cm$^{-1}$ [5]. In metavauxite, the bond valence sum resulting from the structure refinement (Table 5) indicates that the O5–H51 hydroxyl group is involved in a weak but significant hydrogen bridge with the next-nearest-neighbor O2 oxygen connecting the Al1 octahedron with the P1 tetrahedron (Figure 7).

In such a case, the O–H bond strength is lowered, and the corresponding vibrational mode is shifted by~120 cm$^{-1}$ toward lower wavenumbers. This assignment is also consistent with the strong pleochroism observed for the Raman spectra in the O–H stretching region. Inspection of Figure 3a shows that the elongation of the crystal plate must coincide with the **c** crystallographic direction, i.e., with the direction of the octahedral chains. This being the case, the spectrum of Figure 6a (top) has been collected with the E electric vector of the incident beam closely parallel to the O5–H5 bond direction (Figure 3a, see also Figure 7), causing the significant enhancement of the intensity of the Raman scattering.

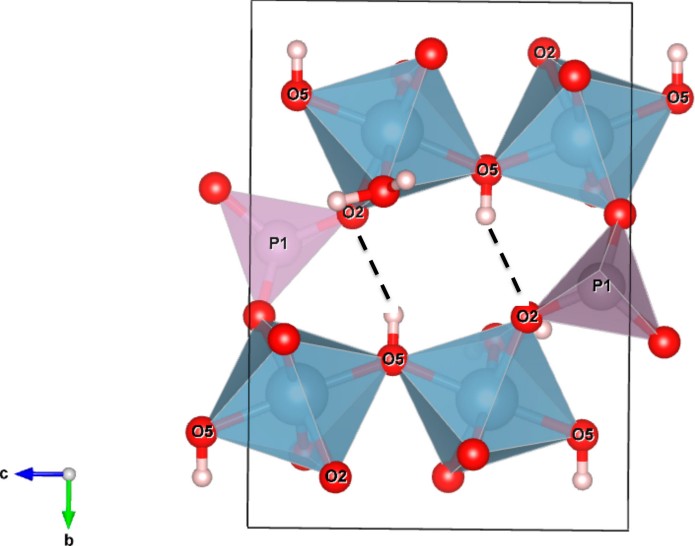

**Figure 7.** The local environment around the OH group in metavauxite viewed along the **a** crystallographic direction; the unit cell is indicated. H51 is involved in a hydrogen bridge (broken line) with the O2 oxygen connecting the Al octahedron (in blue) and the P1 tetrahedron (pink). The O5 ... .O2 donor–acceptor distance is 2.978 Å (Table 4). Figure drawn by VESTA [22].

### 3.3.2. The Low-Frequency Region

The Raman spectrum in the low–frequency 1400–400 cm$^{-1}$ region is given in Figure 8 (top) in comparison with the FTIR powder spectrum (Figure 8, bottom). Measured band positions (wavenumbers, cm$^{-1}$) are listed in Table 6, where the band assignments are based on literature data for similar materials [39–43]. The free phosphate ion, PO$_4^{3-}$ with ideal $T_d$ point symmetry has four modes of vibration: The symmetric stretching $\nu_1(A_1)$ at 980 cm$^{-1}$, the symmetric bending $\nu_2(E)$ at 420 cm$^{-1}$, the asymmetric stretching $\nu_3(F_2)$ at 1082 cm$^{-1}$, and the asymmetric bending $\nu_4(F_2)$ at 567 cm$^{-1}$ (e.g., [39,40]). In accordance with selection rules, the triply degenerate asymmetric stretching and bending modes (F$_2$) are both Raman and infrared active, whereas the non-degenerate symmetric stretching (A$_1$) and the doubly degenerate symmetric bending (E) are Raman active only. In the IR pattern, these latter modes became apparent only when the symmetry of the PO$_4^{3-}$ ion is lowered [39,40]. In such a case, shifts of the absorption bands with respect to the ideal values, band splitting, and appearance of ideally non active IR/Raman modes are observed. Both the FTIR and Raman patterns (Figure 8 and Table 6) suggest that the point symmetry of the PO$_4^{3-}$ group in metavauxite is reduced from $T_d$ to $C_s$, in accordance with the structure refinement that provides four slightly different P1–O distances in the unique P tetrahedron (Table 3). The intense peaks observed in the antisymmetric stretching range at 1142, 1084, and 1049 cm$^{-1}$, respectively (FTIR), and the weak peaks at 1159, 1102, and 1063 cm$^{-1}$ (Raman) can, thus, be assigned to the $\nu_3$ mode.

One component at 992 (Raman) and 997 (IR) cm$^{-1}$ occur as symmetric stretching mode. The antisymmetric bending modes ($\nu_4$, Table 6) are intense in the FTIR spectrum (in the 600–450 cm$^{-1}$ range), while being relatively weak in the Raman pattern.

The $\nu_2(PO_4)^{3-}$ modes, are observed at wavenumbers >400 cm$^{-1}$ (e.g., [2,41]) in the Raman spectrum as medium intense absorptions (Figure 7). Finally, Raman peaks that are found below 200 cm$^{-1}$ are generally classified as lattice modes (e.g., [44]).

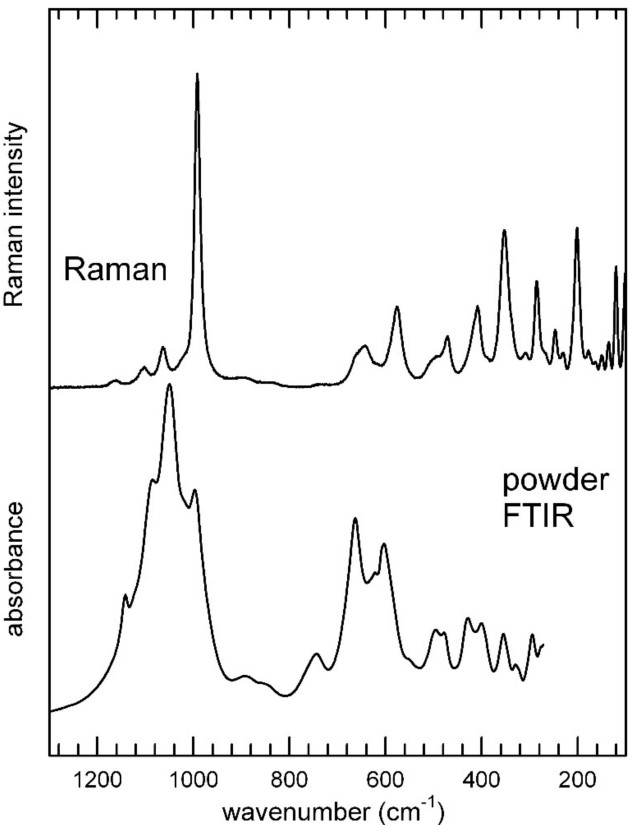

**Figure 8.** Comparison between the Raman (top) and powder FTIR (bottom) spectra of metavauxite in the framework mode region (<1200 cm$^{-1}$).

## 4. Conclusions

Metavauxite, ideally $FeAl_2(PO_4)_2(OH)_2 \cdot 8H_2O$, from Llallagua (Bolivia), was subject to a crystal-chemical investigation, by means of a multi-methodological approach based on EDS microprobe analysis, single crystal X-ray diffraction, and vibrational spectroscopies (Raman and FTIR). The structure was refined in the monoclinic $P2_1/c$ space group, with the following unit cell constants: $a = 10.2449(5)$, $b = 9.5867(5)$, $c = 6.9626(3)$ Å, $\beta = 97.889(3)°$, and $V = 677.36(6)$ Å$^3$. The complex H-bonding scheme in the metavauxite structure is now well defined and nine independent H bonds, with energetically favorable bonding configuration, are described, enabling the understanding of the complex framework consisting of layers along **a** made up of $[Al(PO_4)_3(OH)(H_2O)_2]$ units and isolated $Fe(H_2O)_6$ octahedra, joined together also via the strong hydrogen bonds. FTIR and Raman spectra show, in the $H_2O$ stretching region, a broad absorption consisting of several overlapping components, pertinent $H_2O$ molecules, and the OH groups.

**Author Contributions:** All authors conceived and designed the main ideas, and supervised the whole work; G.C. contributed in EDS analysis, F.C. and G.V. contributed in the single crystal X-ray analysis; G.D.V. contributed in the FTIR analysis; A.M. and G.V. contributed for the Raman analysis. All authors contributed to the redaction of the text.

**Funding:** This research received no external funding.

**Acknowledgments:** The research work has been partly supported by Roma Tre University (Grant to Department of Science, MIUR-Italy Dipartimenti di Eccellenza, ARTICOLO 1, COMMI 314–337 LEGGE 232/2016), by University of Bari (Bari, Italy), PONa3_00369 "Laboratorio per lo Sviluppo Integrato delle Scienze e delle TEcnologie dei Materiali Avanzati e per dispositivi innovativi (SISTEMA)", and by CNR-National Research Council (Rome, Italy).

**Conflicts of Interest:** The authors declare no conflicts of interest. The founding sponsors had no role in the design of the study; in the collection, analyses, or interpretation of data; in the writing of the manuscript, and in the decision to publish the results.

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
