# Peer review of "X-ray Structure Refinement and Vibrational Spectroscopy of Metavauxite FeAl2(PO4)2(OH)2·8H2O"

_crystals, doi:10.3390/cryst9060297_

Round 1
Reviewer 1 Report
This is a good structural job. The data obtained by different methods are consistent with each other. To localize hydrogen atoms from an X-ray diffraction experiment is a very big problem. The authors managed to solve it and to build a network of hydrogen bonds. However, there are some questions and comments.
1. The authors write than the final stages of the refinement did not indicate the need of an extinction correction. How? What is the procedure to detect the need of an extinction correction?
2. How did the authors determine the valence of iron?
3. The section 3 should be titled Results and discussion.
4. Authors should read and refer to the following papers:
Scholz, R, Frost, RL Frota, L, Belotti, FM, Lopez, A. SPECTROCHIMICA ACTA PART A-MOLECULAR AND BIOMOLECULAR SPECTROSCOPY. 2015. V. 151. P. 149-155. Doi 10.1016/j.saa.2015.06.062
Meisser, N, Brugger, J, Krivovichev, S, Armbruster, T, Favreau, G. EUROPEAN JOURNAL OF MINERALOGY. 2012. V. 24. Iss. 4. P. 717-726. Doi 10.1127/0935-1221/2012/0024-2172
The crystal structure of maghrebite, MgAl2(AsO4)2(OH)2·8H2O, is interesting and should be compared with the structure of metavauxite.
5. Wyckoff is written in y, not in i.
6. In fig. 6, I think it is better to remove the frames around the pictures.
7. Table 6 – there is an unknown symbol before Infrared.
Author Response
REFEREE 1
This is a good structural job. The data obtained by different methods are consistent with each other. To localize hydrogen atoms from an X-ray diffraction experiment is a very big problem. The authors managed to solve it and to build a network of hydrogen bonds. However, there are some questions and comments.
1. The authors write than the final stages of the refinement did not indicate the need of an extinction correction. How? What is the procedure to detect the need of an extinction correction?
We used a reflection analysis tool implemented in the refinement software; the tool did not suggest to refine the extinction coefficient.
2. How did the authors determine the valence of iron?
We could not determine directly the Fe oxidation state, because we had no possibility to run a Mössbauer spectrum. However, the Fe valence is relatively well constrained by the stoichiometry; the EDS data provide an Al content ending up to full occupancy of the Al-site, thus Fe must be virtually all ferrous for electroneutrality constraints. Moreover Fe-O bond distances agree well with the estimated one using bond valence methods for 6-coordinated Fe3+.
3. The section 3 should be titled Results and discussion.
OK, corrected
4. Authors should read and refer to the following papers:
Scholz, R, Frost, RL Frota, L, Belotti, FM, Lopez, A. SPECTROCHIMICA ACTA PART A-MOLECULAR AND BIOMOLECULAR SPECTROSCOPY. 2015. V. 151. P. 149-155. Doi 10.1016/j.saa.2015.06.062
Meisser, N, Brugger, J, Krivovichev, S, Armbruster, T, Favreau, G. EUROPEAN JOURNAL OF MINERALOGY. 2012. V. 24. Iss. 4. P. 717-726. Doi 10.1127/0935-1221/2012/0024-2172
The crystal structure of maghrebite, MgAl2(AsO4)2(OH)2·8H2O, is interesting and should be compared with the structure of metavauxite.
In “Related phase” section we did not compare the maghrebite and metavauxite structure because of marked topological differences. Indeed, the maghrebite structure consists of chains of trans-corner-sharing Al octahedra interlinked by PO4 tetrahedra where Al-octahedra are either 5-connected (linked to two octahedra and three PO4 tetrahedra) or 4-connected (linked to two octahedra and two tetrahedra). Differently, in the chains of metavauxite structure the Al-octahedra are all 5-connected (linked to two octahedra and three tetrahedra).
Suggested reference Scholz et al. added.
5. Wyckoff is written in y, not in i.
OK, sorry corrected
6. In fig. 6, I think it is better to remove the frames around the pictures.
OK, done
7. Table 6 – there is an unknown symbol before Infrared.
OK, sorry corrected
Reviewer 2 Report
The paper reviewed is devoted to the detailed description of crystal structure and spectroscopy characteristics. It is an interesting work, but I think that the crystal chemical section should be improved before the publication.
- The first and the main suggestion I want to mention is the total absence of the structural information on the metavauxite varieties (vauxite and paravauxite). I believe that some comparison should be definitely given in the “Related phases” section.
- Lines 25, 81, etc.: “H2O molecules” should be given instead of “water molecules”.
- Line 87: Missed “1” subscript symbol for R value, wrong wR2 value and “2” should be a subscript.
- Line 96: Erroneous “and” in the “Crystal and structure refinement”.
- Line 122: “Powder FTIR spectra …” – how the phase purity was checked?
- Lines 144-145: “Formula was calculated on the basis of 9 negative charges” – smth wrong here: O8(OH)2 will give the charge of -18.
- Line 156: It’s not a Fe-O, it is Fe-H2O contact.
- Lines 157-158, Table 3: It’s better to specify for clarity the contacts (O, OH or H2O): for instance, Al1-O5H, Al1-H2O6, etc.
Author Response
REFEREE 2
The paper reviewed is devoted to the detailed description of crystal structure and spectroscopy characteristics. It is an interesting work, but I think that the crystal chemical section should be improved before the publication.
- The first and the main suggestion I want to mention is the total absence of the structural information on the metavauxite varieties (vauxite and paravauxite). I believe that some comparison should be definitely given in the “Related phases” section.
In the revised manuscript structural differences among metavauxite varieties are shortly discussed.
- Lines 25, 81, etc.: “H2O molecules” should be given instead of “water molecules”.
OK, corrected
- Line 87: Missed “1” subscript symbol for R value, wrong wR2 value and “2” should be a subscript.
OK, corrected
- Line 96: Erroneous “and” in the “Crystal and structure refinement”.
OK, corrected
- Line 122: “Powder FTIR spectra …” – how the phase purity was checked?
The pellet was done with a very small and optically pure single-crystal manually separated from the specimen and checked by X-ray diffraction.
- Lines 144-145: “Formula was calculated on the basis of 9 negative charges” – smth wrong here: O8(OH)2 will give the charge of -18.
Sorry, the formula was calculated on the basis of 9 oxygen atoms (18 negative charges).
- Line 156: It’s not a Fe-O, it is Fe-H2O contact.
OK we indicated as ‘Fe-H2O bond distances’ all along the paragraph.
- Lines 157-158, Table 3: It’s better to specify for clarity the contacts (O, OH or H2O): for instance, Al1-O5H, Al1-H2O6, etc.
OK, changed
Reviewer 3 Report
The article by Della Ventura et al. details a thorough structural and spectroscopic study on the Metavauxite, whose structural details lacked a accurate description, in particular regarding the network of the hydrogen bonds within this crystal phase. The accurate work carried out by the authors can be of interest to researchers working in this geological and/or crystallographic field. Some minor revisions are recommended to make this manuscript suitable for publication in Crystals. These recommendations are listed as follows (the starting number clarifies the line in the manuscript where the correction should be made):
19- the units are lacking their formal charge, and the iron complex is cationic and should be reported appropriately.
28- A typo is present: PO4 -3 to be corrected in PO4 3-
83- The authors should state that the short O-H distances are due to the impossibility to locate the hydrogen positions with X rays, since it is possible only to guess their orientation by the electron density maps. For this reason the positions of Hydrogens from XRD experiments should always be restrained when not determined with other methods sensitive to the protons' positions such as, for instance, neutron diffraction.
Figure 3- caption: I would suggest to use “along the a crystallographic axis” and “along the c crystallographic axis” instead of the [u,v,w] notation as it can be confused with the [h,k,l] notation (if not specified). This is important since in this case the [100] and [001] directions differ depending on which notation is intended. The same comment is valid for the caption of Figure 7.
214: the ions are also in thsis case reported without formal charges.
Figure 5: the resolution of the image should definitely be improved.
276: the link should be reported as reference and not in the text for the sake of a fluent reading.
332: the title of the paragraph has no listing number (I suppose should be 3.2.2).
Moreover, the Checkcif test performed on the cif fle presents three "alert A" due to missing information in the cif file. These information should definitely be added to the cif file prior to its deposition, in agreement with the guidelines of the International Union of Crystallography.
Author Response
REFEREE 3
The article by Della Ventura et al. details a thorough structural and spectroscopic study on the Metavauxite, whose structural details lacked a accurate description, in particular regarding the network of the hydrogen bonds within this crystal phase. The accurate work carried out by the authors can be of interest to researchers working in this geological and/or crystallographic field. Some minor revisions are recommended to make this manuscript suitable for publication in Crystals. These recommendations are listed as follows (the starting number clarifies the line in the manuscript where the correction should be made):
19- the units are lacking their formal charge, and the iron complex is cationic and should be reported appropriately.
OK, corrected
28- A typo is present: PO4 -3 to be corrected in PO4 3-
OK, corrected
83- The authors should state that the short O-H distances are due to the impossibility to locate the hydrogen positions with X rays, since it is possible only to guess their orientation by the electron density maps. For this reason the positions of Hydrogens from XRD experiments should always be restrained when not determined with other methods sensitive to the protons' positions such as, for instance, neutron diffraction.
OK, thanks, this point has been clarified and completed in the text.
Figure 3- caption: I would suggest to use “along the a crystallographic axis” and “along the c crystallographic axis” instead of the [u,v,w] notation as it can be confused with the [h,k,l] notation (if not specified). This is important since in this case the [100] and [001] directions differ depending on which notation is intended. The same comment is valid for the caption of Figure 7.
OK, corrected.
214: the ions are also in this case reported without formal charges.
OK, corrected
Figure 5: the resolution of the image should definitely be improved.
Ok, we improved as possible the resolution of Figure 5.
276: the link should be reported as reference and not in the text for the sake of a fluent reading.
OK, we used here the way these kind of reference is cited in Minerals, another MDPI journal:
..... (see for example the pertinent mindat.org database entry [37])
332: the title of the paragraph has no listing number (I suppose should be 3.2.2).
Sorry, corrected
Moreover, the Checkcif test performed on the cif fle presents three "alert A" due to missing information in the cif file. These information should definitely be added to the cif file prior to its deposition, in agreement with the guidelines of the International Union of Crystallography.
Ok, we revised the cif file with missing information, and uploaded it with the name checkcif. We will send the revised version of ICSD once the paper will be published, thus complete of full reference.
Reviewer 4 Report
The authors present the crystal structure of metavauxite with more accurate hydrogen positions obtained by single crystal X-ray diffraction, and Raman and FTIR analysis of the mineral.
Line 16:
Please, define FTIR and (if it is allowed to have references in the abstract, add references for “from previous literature data” in here.
Line 36:
Please, check the chemical composition given for vauxite. I believe the number of water molecules should be different than the given here.
Line 41:
What do the authors mean by “standard information”? Crystallographic information?
Line 44:
What do the authors mean by “modern X-ray data”?
Lines 51 and 60:
If there is water loss, did the authors observe any type of disorder? Also, maybe the structure is already different due to some transition caused by water loss? Did the authors try measuring the crystal at a low temperature?
Line 68:
One might remove the definition for CCD, as it is quite well known.
More important here would be to mention which detector the authors used. PHOTON II, III?
Line 70:
Were there more than a unit cell determination? If so, which others were found?
Table 1:
Could the authors discuss how significant is the difference peak in the case of the hydrogen positions in the structure? Maybe add an image (for example, from VESTA) showing the difference density.
Line 122 and 123:
Please, either use the abbreviation and then the meaning, or the opposite. For example, lines 122 “MIR (medium infrared range) and 130 “charge-coupled device (CCD)”.
Figure 2:
There is an “h” floating on the image.
Figure 4:
It would be helpful to have an image with clearer text. The names of the minerals are quite fuzzy.
Table 5:
Just a mistyping, I believe, but the O4-P1 bond-valence should be 1.242.
Line 327:
Is the elongation of the crystal plate along the c axis visible through the atomic displacement parameters?
Author Response
REFEREE 4
The authors present the crystal structure of metavauxite with more accurate hydrogen positions obtained by single crystal X-ray diffraction, and Raman and FTIR analysis of the mineral.
Line 16:
Please, define FTIR and (if it is allowed to have references in the abstract,
OK, done
add references for “from previous literature data” in here.
We deleted “from previous literature data” in the abstract. Previous literature is quoted in Introduction.
Line 36:
Please, check the chemical composition given for vauxite. I believe the number of water molecules should be different than the given here.
OK, sorry corrected
Line 41:
What do the authors mean by “standard information”? Crystallographic information?
OK, clarified
Line 44:
What do the authors mean by “modern X-ray data”?
data collected with modern instrument... corrected
Lines 51 and 60:
If there is water loss, did the authors observe any type of disorder? Also, maybe the structure is already different due to some transition caused by water loss? Did the authors try measuring the crystal at a low temperature?
No, we could not collect low T X-ray data. However, differently from vauxite and paravauxite, in metavauxite all water molecules are coordinated by cations. Therefore, these water molecules are strongly bound in the structure and cannot be easily removed from the structure. Moreover, EDX results agree well the ideal stoichiometry implying no H2O deficiencies with respect to the ideal composition. We didn’t observe any type of disorder.
Line 68:
One might remove the definition for CCD, as it is quite well known.
More important here would be to mention which detector the authors used. PHOTON II, III?
OK, corrected
Line 70:
Were there more than a unit cell determination? If so, which others were found?
Just one initial cell determination has been found. Corrected.
Table 1:
Could the authors discuss how significant is the difference peak in the case of the hydrogen positions in the structure? Maybe add an image (for example, from VESTA) showing the difference density.
The hydrogen positions correspond to the most intense (and then most significant) peaks found in the difference Fourier map. These peaks are given in the new cif file and below.
# Highest difference peak 1.114,
# deepest hole -0.756, 1-sigma level 0.176
# (before H assignment)
# Q1 1 0.4938 0.4067 0.3700 11.00000 0.05 1.11
# Q2 1 0.1016 0.2874 0.3582 11.00000 0.05 1.04
# Q3 1 0.7766 0.3354 0.4843 11.00000 0.05 0.92
# Q4 1 0.0770 0.0618 0.3514 11.00000 0.05 0.86
# Q5 1 0.3408 0.4590 0.0566 11.00000 0.05 0.86
# Q6 1 0.2780 0.1603 0.1026 11.00000 0.05 0.86
# Q7 1 -0.0393 0.0610 0.0505 11.00000 0.05 0.82
# Q8 1 0.1491 0.3798 0.2605 11.00000 0.05 0.82
# Q9 1 0.3715 0.4871 0.0300 11.00000 0.05 0.78
# Q10 1 -0.0433 0.0919 0.3270 11.00000 0.05 0.75
# Q11 1 0.3368 0.1131 0.2961 11.00000 0.05 0.75
# Q12 1 0.3629 0.6037 0.0373 11.00000 0.05 0.65
# Q13 1 0.2626 0.5017 0.0885 11.00000 0.05 0.65
# Q14 1 0.7838 0.3880 0.3305 11.00000 0.05 0.62
# Q15 1 -0.0104 0.1296 0.2863 11.00000 0.05 0.56
# Q16 1 0.0517 0.2482 0.3396 11.00000 0.05 0.55
# Q17 1 -0.0651 0.0831 0.3891 11.00000 0.05 0.55
# Q18 1 0.1846 0.3209 0.3237 11.00000 0.05 0.55
# Q19 1 0.3317 0.1996 0.1680 11.00000 0.05 0.55
# Q20 1 0.1670 0.3713 0.3447 11.00000 0.05 0.54
Line 122 and 123:
Please, either use the abbreviation and then the meaning, or the opposite. For example, lines 122 “MIR (medium infrared range) and 130 “charge-coupled device (CCD)”.
OK, corrected
Figure 2: There is an “h” floating on the image.
We replaced Figure 2 with a new figure.
Figure 4:
It would be helpful to have an image with clearer text. The names of the minerals are quite fuzzy.
OK, we made clearer the Figure 4.
Table 5:
Just a mistyping, I believe, but the O4-P1 bond-valence should be 1.242.
OK, corrected, sorry
Line 327:
Is the elongation of the crystal plate along the c axis visible through the atomic displacement parameters?
In the uploaded cif file the atomic displacement parameters show U11 or U22 values slightly higher than U33. This is also displayed in the displacement ellipsoid plot in the new Figure 2.